COMMENT

# AuthorArranger automates formatting title pages and author affiliations for manuscript submissions

Mitchell J. Machiela [1✉] & Geoffrey Tobias [1]

Scientific investigations are becoming increasingly collaborative resulting in growing lists of contributors to studies that can be time consuming to organize when formatting journal title pages. Here we present AuthorArranger, a freely available web tool to assist with automating journal title page creation from imported author details.

How many scientists does it take to perform a study? Peer-reviewed published articles have author lists ranging from 1 author[1] up to a staggering 5154 authors[2], with one notable trend: scientific investigations have become increasingly collaborative. In fact, from 2014–2019 the number of published manuscripts listing over 1000 coauthors is over 1300 and this trend in hyper-authorship is expected to grow[3]. Contributors to studies often span multiple areas of expertise, involve numerous scientific institutions, and include several regions around the globe. The remote, teleworking environment necessitated by the COVID-19 pandemic has further facilitated collaboration by promoting mass adoption of virtual meeting platforms that enable (almost) seamless meetings of individuals from around the world as easily as meeting with a colleague within your department. The increased collaborative environment in scientific investigations has augmented sample sizes, assisted in breaking down research silos, and helped pave the way for interdisciplinary, multi-institutional and integrative studies that address critical gaps in understanding and accelerate scientific progress.

## Comment

The problem: classical methods of submitting manuscripts for peer-reviewed publication have not kept pace with the large author lists needed to acknowledge the expanding number of scientists contributing to these efforts. The result is a cumbersome submission process. First, a formatted journal title page needs to be generated that correctly details author names, order of contribution and affiliations. Second, author contributions need to be detailed that match efforts and responsibilities to each author's name, usually abbreviated with their initials. Finally, electronic forms on journal submission portals need to be completed with contributor's details. In studies with limited author lists these tasks are not a heavy lift, but for large collaborative efforts with expansive author lists these tasks can amount to a substantial investment of time and effort; especially when no automated approach exists to perform the task.

We present to the scientific community a freely available online web tool to assist with automating the first step of formatting a title page for manuscript submission (Fig. 1). AuthorArranger (https://authorarranger.nci.nih.gov/) is designed to rapidly format journal title pages using an uploaded spreadsheet of author details, enabling researchers to conquer title pages in a matter of seconds—regardless of the size of author lists. Features include the ability to upload author details; flexible formats for matching journal style; ability to add, reorder or

[1]Division of Cancer Epidemiology and Genetics, National Cancer Institute, Rockville, MD, USA. ✉email: mitchell.machiela@nih.gov

**Fig. 1 AuthorArranger screenshot.** Screenshot of AuthorArranger demonstrating features and flexibility of the free and publicly available online platform to conquer journal title pages in seconds.

remove authors; instantaneous preview of format; and download of .docx files for adding to manuscripts. Authors only need to download the provided AuthorArranger template and upload a completed version of author details to the site. AuthorArranger rapidly takes care of all the formatting.

## Outlook
AuthorArranger represents a new and simplified approach for generating and submitting manuscripts and supporting materials to scientific journals for rapid dissemination of research findings. As technology and web interfaces continue to evolve, our hope is that additional tools like AuthorArranger continue to simplify and speed up the process of submitting collaborative manuscripts with large lists of contributors.

**Reporting summary**. Further information on research design is available in the Nature Research Reporting Summary linked to this article.

## Data availability
AuthorArranger is available online at https://authorarranger.nci.nih.gov/#/.

## Code availability
All development code for AuthorArranger is available at https://github.com/CBIIT/nci-webtools-dceg-author-arranger.

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

## Acknowledgements
AuthorArranger was created in collaboration with the National Cancer Institute (NCI) Center for Biomedical Informatics and Information (CBIIT). We thank Brian Park, Ye Wu, Sue Pan and Mei Liu from the NCI CBIIT for their technical support in developing this tool and Wendy Schneider-Levinson from the NCI Division of Cancer Epidemiology and Genetics (DCEG) for assistance with Section 508 compliance and usability testing. Funding for AuthorArranger comes from the 2018 NCI DCEG Informatic Tool Challenge. All AuthorArranger support is provided by the Intramural Research Program of the NCI for a minimum of two years.

## Author contributions
M.J.M. Conceived the project, secured funding, supervised the work, drafted the manuscript. G.T. Conceived the project, secured funding, supervised the work, drafted the manuscript.

## Funding

## Competing interests

The authors declare no competing interests.
