## [Peer Review file · Communications Biology]

REVIEWER COMMENTS:

Reviewer #1 (Remarks to the Author):

While not promising to change the world, the AuthorArranger tool is potentially quite useful, and draws attention to a very reasonable issue.

The only comment which I would have on the manuscript would be to perhaps ever so slightly more thoroughly describe\comment on the quantitative characteristics of 'hyper-authorship'. It is not, at present, really clear at all what the Lyon citation is doing other than providing an example of a solo-authored paper. There does exist a huge body of scientometric work which examines collaboration, and some work which looks at hyper-authorship in particular which could provide some headline summary statistics. See, for example, this Clarivate Global Research Report: "Multi-authorship and research analytics" which was covered by a Nature News article on 13 December 2019 (<https://doi.org/10.1038/d41586-019-03862-0>).

I've tried the app, and while I haven't tried to find every edge case which might cause issues, it seems to work flawlessly. Commendations to the authors for this contribution to the scientific community!

Reviewer #2 (Remarks to the Author):

In the manuscript, authors Machiela and Tobias an online web tool (AuthorArranger) to assist with automating the formatting of a title page for manuscript submission. As they point out, long authorship lists can be burdensome to populate and modify, and their tool is equipped facilitate the addition of authors and affiliations and to adapt to a number of formats. This tool is likely to be embraced by investigators across disciplines, particularly those from fields with long authorship lists. In the future, it would be great if such a tool could also incorporate authorship contributions, which are increasingly part of the manuscript submission process. The manuscript is to the point and I don't have any additional comments.

Reviewer #1 (Remarks to the Author):

While not promising to change the world, the AuthorArranger tool is potentially quite useful, and draws attention to a very reasonable issue.

Thanks for your time reviewing AuthorArranger and for highlighting that formatting journal title pages is an important issue.

The only comment which I would have on the manuscript would be to perhaps ever so slightly more thoroughly describe\comment on the quantitative characteristics of 'hyper-authorship'. It is not, at present, really clear at all what the Lyon citation is doing other than providing an example of a solo-authored paper. There does exist a huge body of scientometric work which examines collaboration, and some work which looks at hyper-authorship in particular which could provide some headline summary statistics. See, for example, this Clarivate Global Research Report: "Multi-authorship and research analytics" which was covered by a Nature News article on 13 December 2019 (<https://doi.org/10.1038/d41586-019-03862-0>).

Thanks for this helpful comment. We have revised the text to clarify the Lyon reference is an example of a single author paper. As Reviewer 1 suggested, we also slightly expanded on quantitative characteristics of hyper-authorship and included the recommended citation.

I've tried the app, and while I haven't tried to find every edge case which might cause issues, it seems to work flawlessly. Commendations to the authors for this contribution to the scientific community!

Glad all ran smoothly with your initial use of the app. Thanks for sharing your positive experience.

Reviewer #2 (Remarks to the Author):

In the manuscript, authors Machiela and Tobias an online web tool (AuthorArranger) to assist with automating the formatting of a title page for manuscript submission. As they point out, long authorship lists can be burdensome to populate and modify, and their tool is equipped facilitate the addition of authors and affiliations and to adapt to a number of formats. This tool is likely to be embraced by investigators across disciplines,

particularly those from fields with long authorship lists. In the future, it would be great if such a tool could also incorporate authorship contributions, which are increasingly part of the manuscript submission process. The manuscript is to the point and I don't have any additional comments.

We thank Reviewer 2 for the positive comments and agree AuthorArranger will likely be embraced by investigators across disciplines. We hope to add additional functionality to the tool including the suggested authorship contributions, but have limited resources at the moment to implement these changes.